# Sagnac Interferometric Temperature Sensor Based on Boron-Doped Polarization-Maintaining Photonic Crystal Fibers

**Lan Cheng** [1], **Jun Liang** [1], **Shiwei Xie** [2] and **Yilin Tong** [1,*]

1   Information Science and Technology Department, Wenhua College, Wuhan 430074, China
2   School of Urban Construction, Wuhan University of Science and Technology, Wuhan 430065, China
*   Correspondence: t111l@126.com

**Abstract:** A sensitive temperature sensor was demonstrated using boron-doped polarization-maintaining photonic crystal fiber (PM-PCF) as a Sagnac interferometer (SI). This boron-doped PM-PCF combines both the geometric birefringence introduced by the PCF structure design and the stress birefringence introduced by the boron-doped stress-applying parts. However, we found that the stress birefringence dominates the total birefringence of the sensor by numerical analysis. In the experiments, the fabricated sensor exhibited the highest temperature sensitivity of −1.83 nm/°C within the wide temperature range of 28~76 °C. The temperature sensitivity was mainly derived from the stress birefringence of boron-doped PM-PCF SI. These findings provide some support for the designation of high-precision temperature sensors.

**Keywords:** polarization maintaining fiber; photonic crystal fiber; temperature sensing; Sagnac interferometer





## 1. Introduction

Interferometric optical fiber temperature sensors have gained widespread attention for their inherent advantages of strong anti-interference capability, small size, lightweight, and high sensitivity. Polarization-maintaining fibers (PMFs) are widely used in Sagnac interferometers (SIs) due to their excellent polarization properties [1,2]. Conventional PMFs (e.g., panda type and bowtie type) have a much higher thermal expansion coefficient than pure silica cladding due to boron-doped stress-applying parts (SAPs), which are very sensitive to temperature [3]. However, the conventional PMF with a simple and solid structure is hardly modified for diverse temperature sensing demands.

Photonic Crystal Fibers (PCFs) [4–6] are a kind of fiber with a flexible structural design. By arranging the geometry or distribution of the core and cladding, PCFs can obtain ultrahigh birefringence [7,8]. High birefringence PCFs are good choices for compact fiber SIs. Although this kind of PM-PCFs composed of a single silica glass material has relatively high birefringence, the low thermal expansion coefficient and thermo-optic coefficient make PM-PCFs SIs sensitive to low temperatures [9]. In order to improve the temperature sensitivity, boron-doped SAPs were added during the fabrication of PM-PCFs. This boron-doped PM-PCF combines both the geometric birefringence introduced by the PCF structure design and the stress birefringence introduced by the boron-doped SAPs. Crystal Fibre in Denmark and Jena University in Germany have produced this kind of boron-doped PM-PCF and double-clad ytterbium-doped PM-PCF [10,11], but their sensing properties have not been studied.

In this paper, a boron-doped PM-PCF was designed according to SI and fabricated by micro-processing on an optical fiber fusion splicer. The temperature response of this SI was studied by numerical analysis and temperature sensing experiments. The introduction of boron-doped SAPs into the PM-PCF greatly elevates the temperature sensitivity of SI. To inspire future sensor designs, we also elucidate the sensing mechanism of the boron-doped PM-PCF SI.

## 2. Fiber Structure and Experimental Setup

Figure 1a shows the optical microscope image of the PM-PCF. The PM-PCF was made by the standard stack-and-draw process as follows: (1) the silica glass tubes, silica glass rods and boron-doped silica glass rods were straightened into capillary tubes, capillary rods and boron-doped capillary rods with diameters of about 1 mm, respectively; (2) the capillary tubes were stacked by hand into a hexagonal structure. After stacking, the fiber core was replaced by 7 capillary rods, and each boron-doped region was replaced by 19 boron capillary rods; (3) this preform inserted into the jacketing tube is moved into the high-temperature furnace of a drawing tower and then fused together and drawn down to the final PM-PCF. The fiber diameter and core diameter are 115 and 9.2 μm, respectively. Two fan-shaped areas on the x-axis have an air hole pitch (Λ) of 5.0 μm and a 66% ratio (d/Λ) of hole diameter to pitch. Two other fan-shaped areas on the y-axis are formed by replacing air holes with boron-doped rods.

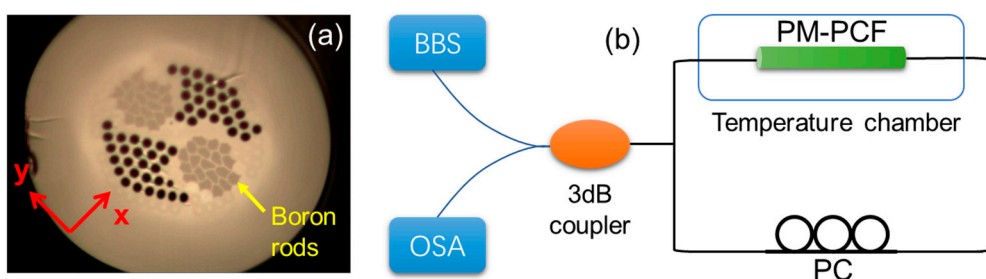

**Figure 1.** (**a**) Optical micrograph of fiber ends face, and (**b**) schematic diagram of the SI experimental setup.

The SI setup used in the experiment is shown in Figure 1b, which includes a broadband source (BBS, 1250–1650 nm), an optical spectrum analyzer (OSA, YOKOGAWA AQ6370D), a 3-dB coupler, a polarization controller (PC), a temperature chamber, and a PM-PCF. The PC is used to adjust the polarization state of the incident light. The PM-PCFs were placed in a temperature chamber with an accuracy of 0.1 °C. To obtain a precise temperature control, the temperature chamber was heated to the highest temperature (100 °C) that the temperature chamber could bear, and then naturally cooled. Natural cooling can lead to a slowly decreasing temperature around the PM-PCFs. During the experiments, the PM-PCFs were kept straight in the temperature chamber to prevent bending and torsion. The light from the broadband light source is split into two beams by a 3-dB coupler. Inside the fiber optic ring, one beam travels clockwise, and the other travels counterclockwise. Due to the existence of PM-PCF birefringence, there is an optical path difference between the two beams of light during the propagation process, and the interference spectrum can be detected by the optical spectrum analyzer. The transmission spectrum (*T*) of a SI can be approximated as a periodic function of wavelength, namely

$$T = (1 - \cos(2\pi BL/\lambda))/2 \tag{1}$$

where $B = |n_x - n_y|$ is the phase birefringence of the PM-PCF, *L* is the length of the PM-PCF in the fiber loop, and $\lambda$ is the working wavelength. When the wavelength satisfies the phase condition ($\frac{2\pi BL}{\lambda} = 2m\pi$, where *m* is a positive integer), a trough appears in the interference spectrum. The wavelength of m-order interference dip was calculated as:

$$\lambda_m = B(\lambda)L/m \tag{2}$$

The wavelength spacing between adjacent interference dips, namely the free spectral range (*FSR*), can be deduced:

$$FSR = \lambda^2/BL \tag{3}$$

By regulating the ambient temperature around the PM-PCF, the phase birefringence and the fiber length were changed, which could lead to a shift of the interference dip in the interference spectrum. Thus, the temperature variation can be characterized by the shift of the interference dip. The temperature sensitivity can be derived by deriving the temperature $T$ on both sides of the trough wavelength formula:

$$S(T) = \frac{d\lambda_m}{dT} = \frac{\left(\frac{\partial B(\lambda,T)}{\partial T} + B(\lambda,T) * \alpha\right) * \lambda_m(T)}{B_g(\lambda,T)} \tag{4}$$

where $\alpha = (1/L)dL/dT$ is the thermal expansion coefficient of the fiber, and $B_g = B(\lambda,T) - \lambda * \partial B(\lambda,T)/\partial\lambda$ is the real-time group birefringence. It can be known from the formula that the temperature sensitivity of SI is related to the wavelength of the interference dip, the thermal expansion coefficient of the fiber, the phase birefringence, the group birefringence, and the rate of change in the phase birefringence with temperature.

## 3. Numerical Analysis of Temperature Characteristics

To theoretically explore the mechanism of the SI sensor, it is necessary to calculate the birefringence of the optical fiber first. For this highly birefringent fiber containing SAP, its phase birefringence can be divided into three parts [12]:

$$B = B_G + B_S + B_{S0} \tag{5}$$

Among them, $B_G$ is the birefringence caused by the geometric asymmetry of the core and cladding, $B_S$ is the stress birefringence caused by SAPs, and $B_{S0}$ the self-stress birefringence caused by the thermal expansion difference of the core. Birefringence in boron-containing fibers is created by elastic and hydrostatic mechanisms [13]. Since the core of boron-doped PM-PCF is pure silica glass, the item $B_{S0} = 0$. That is, phase birefringence simplifies to:

$$B = B_G + B_S \tag{6}$$

Geometric birefringence $B_G$ is determined by the refractive index profiles in two orthogonal directions. Stress birefringence is due to the different thermal expansion coefficients between boron-doped rods and pure silica glass. After the preform is softened and moved out from a heating furnace, the temperature will drop rapidly. The temperature variations make the boron-doped region shrink strongly due to the large thermal expansion coefficient. Stress is thus generated in the fiber and maintained, which results in a stress birefringence for the fiber. According to the elastic-optic effect, the relationships between the effective refractive index and the stress are shown as follows:

$$n_x(x,y) = n_{x0} - C_1\sigma_x(x,y) - C_2\left(\sigma_y(x,y) + \sigma_z(x,y)\right) \tag{7}$$

$$n_y(x,y) = n_{y0} - C_1\sigma_y(x,y) - C_2\left(\sigma_x(x,y) + \sigma_z(x,y)\right) \tag{8}$$

where $n_{x0}$ and $n_{y0}$ are the refractive indices without considering the stress, $\sigma_x(x,y)$, $\sigma_y(x,y)$ and $\sigma_z(x,y)$ are the spatial distribution of stress in three vertical directions, $C_1$ and $C_2$ are the elastic-optic coefficients of the optical fiber materials ($C_1 = 4.2 \times 10^{-12}$/Pa, $C_2 = 0.65 \times 10^{-12}$/Pa for silica glass) [14]. Then the average refractive index of the two polarization states can be expressed as:

$$\bar{n}_x = \frac{1}{A}\iint n_x(x,y)dxdy, \tag{9}$$

$$\bar{n}_y = \frac{1}{A}\iint n_y(x,y)dxdy \tag{10}$$

where $A$ is the core area. Thus, the fiber birefringence can be expressed as the refractive index difference between the two polarization states, namely:

$$B_{av} = \overline{n}_x - \overline{n}_y \tag{11}$$

The parameters used in the calculation of fiber stress distribution are: Young's modulus E = 7380 kg/mm$^2$, Poisson's ratio $\upsilon$ = 0.186, boron-doped glass thermal expansion coefficient $\alpha_{BS}$ = 2.6 × 10$^{-6}$/°C, silica glass thermal expansion coefficient $\alpha_S$ = 5.5 × 10$^{-7}$/°C, silica glass softening temperature $T$ = 1000 °C, room temperature $T_r$ = 30 °C. Considering the core-cladding ratio of the boron rod is about 90%, the boron-doped region could be approximated as a polygon in order to simplify the calculation model. Figure 2 is a contour extraction diagram of the end face of the fiber. The fiber stress distribution and transmission modes are calculated using the extracted contour maps.

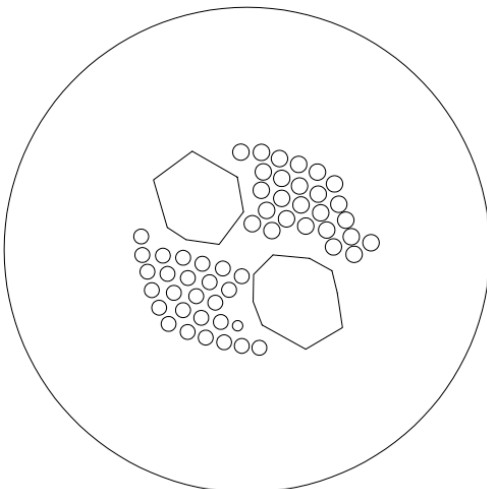

**Figure 2.** The contour extraction map of the fiber end face.

Figure 3 shows the fiber stress distribution and fundamental mode field distribution at 1550 nm and 30 °C. The stress is mainly concentrated in the boron rod region, and the stress direction points to the boron rod region. Different from the circular symmetric fundamental mode field distribution of the traditional single-mode fiber, the fundamental mode field of the fiber is distorted due to the symmetrical destruction of the end structure of the fiber (Figure 3b). The calculated birefringence of the fiber is 2.05 × 10$^{-4}$. In contrast, without regard to the presence of stress, the birefringence of the fiber is 6.88 × 10$^{-5}$. Thus, the geometric birefringence accounts for only one-third of the total birefringence, indicating that the birefringence of the boron-doped PM-PCF is mainly caused by the introduction of SAPs.

Figure 4 shows the modal phase birefringence and modal group birefringence from 1250 to 1650 nm at 30 and 40 °C. The phase birefringence decreases with increasing wavelength. The temperature shows a negative impact on the phase birefringence, which at 30 °C was higher than that at 40 °C. Based on Equation (3), the size of FSR is proportional to the square of the wavelength and inversely proportional to the phase birefringence and the fiber length. For the Sagnac ring with a certain PM-PCF length, since the birefringence of the boron-doped PM-PCF mode decreases with the increase in the wavelength, the FSR increases with the increase in the wavelength. The group birefringence was calculated from the relationship between the phase birefringence and the group birefringence, which is also presented in Figure 4. The group birefringence increases with increasing wavelength, while the group birefringence at 40 °C is lower than that at 30 °C.

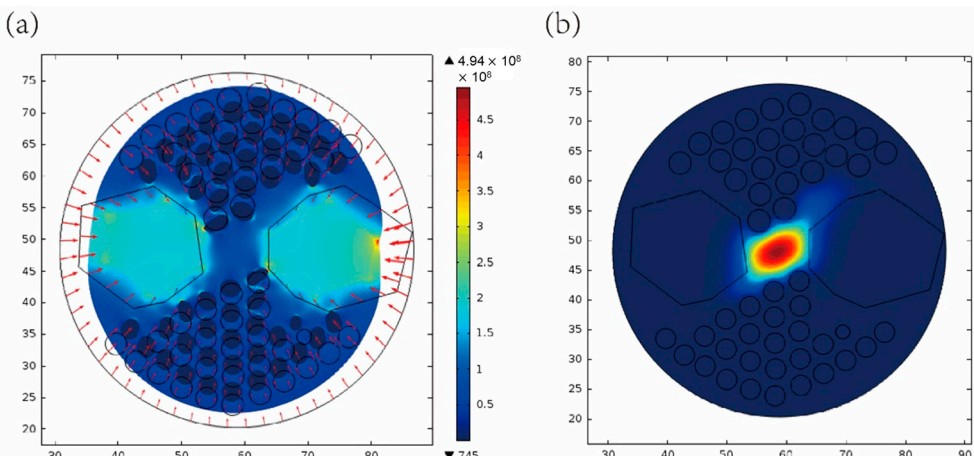

**Figure 3.** (**a**) Stress distribution diagram of PM-PCF. (**b**) Fundamental mode field distribution at 1550 nm.

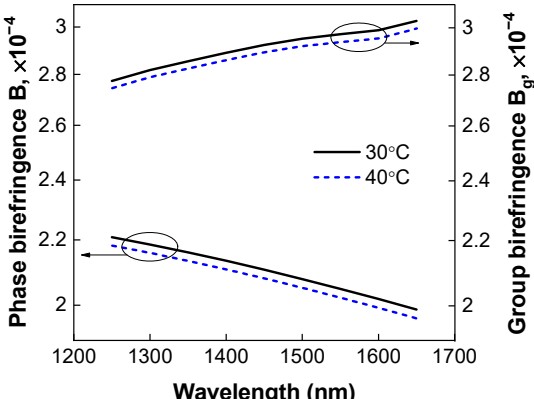

**Figure 4.** Variation of modal phase birefringence B and group birefringence Bg.

By polynomial fitting of the birefringence, the $\partial B(\lambda, T)/\partial T$ at a certain wavelength can be derived. Thus, the values of $\partial B(\lambda, T)/\partial T$ at different wavelengths were obtained by this method. As shown in Figure 5a, the $\partial B(\lambda, T)/\partial T$ at 30 °C were in the magnitude order of $10^{-7}$ °C$^{-1}$, which is about 1000 times of the value of $B(\lambda, T) \times \alpha$. Therefore, the value of $B(\lambda, T) \times \alpha$ in Equation (4) can be ignored, which suggests the movement of the interference dip as the temperature rises is mainly induced by the change in birefringence for the boron-doped PM-PCF SI. As shown in Figure 5b, the calculated temperature sensitivities (S) at 1350, 1450, and 1550 nm were similar around $-1.2$ nm/°C. The negative value means that the interference dip shifts to the short-wave direction as the temperature increases. The S values almost remain stable under different temperatures but tend to be more negative with the wavelength increase from 1350 to 1550 nm. Since the absolute value of the S value at the longer wavelength is larger, interference dip shift would be more obvious. Taken together, these results suggest that there is a linear relation between temperature and wavelength for the boron-doped PM-PCF SI, which enables it to be a favorable sensor of temperature.

Optics

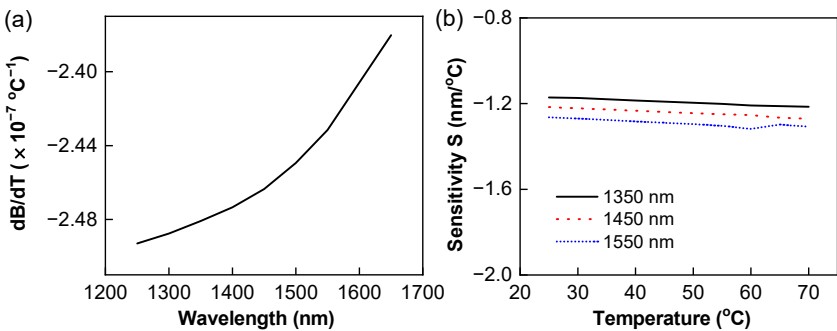

**Figure 5.** (**a**) Variation of $\partial B(\lambda, T)/\partial T$ with wavelength at 30 °C, and (**b**) temperature sensitivity (S) at 28–76 °C.

## 4. Temperature Sensing Experiment

In order to confirm the results obtained by numerical analysis, temperature-sensing experiments were carried out using self-designed PM-PCFs. We firstly examined the impacts of fiber length on the transmission spectra based on both experimental and simulated results (Figure 6). Interference spectrum was both observed for the PCF at lengths of 7.2 and 14.7 cm. Specifically, there are four interference dips in the PCF of 7.2 cm and eight interference dips at 14.7 cm. The deepest interference dip appears at 1553.78 and 1547.68 nm for the fibers of 7.2 and 14.7 cm, respectively. The FSR increases with the wavelength. The simulated transmission spectrum based on the calculated phase birefringence is also presented in Figure 6. The number of interference peaks shows no difference between the experimental and simulated results at the two lengths. We also observed the same trend of FSR with wavelengths from the experimental and simulated spectra.

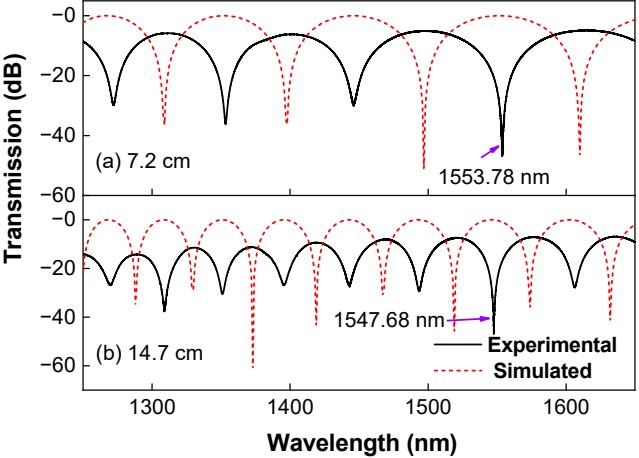

**Figure 6.** Experimental and simulated transmission spectra of boron-doped PM-PCF at lengths of (**a**) 7.2 and (**b**) 14.7 cm.

To explore the impact of fiber length on the temperature sensitivity, we extracted the wavelength of the deepest interference dip from the transmission spectra of the PCF at lengths of 7.2 and 14.7 cm. As shown in Figure 7, the interference dip of different fiber lengths varied with temperature in similar trends, showing parallel fitting lines. The temperature sensitivity could be obtained from the slope values of the fitting line. The fiber of 14.7 cm shows a higher temperature sensitivity than that of 7.2 cm fiber, although the difference is insignificant. In line with our findings, previous work also reported that PMF with different lengths exhibited approximate temperature sensitivities [15]. These findings suggest that the fiber length has little effect on the temperature sensitivity of the boron-doped PM-PCF SI temperature sensor.

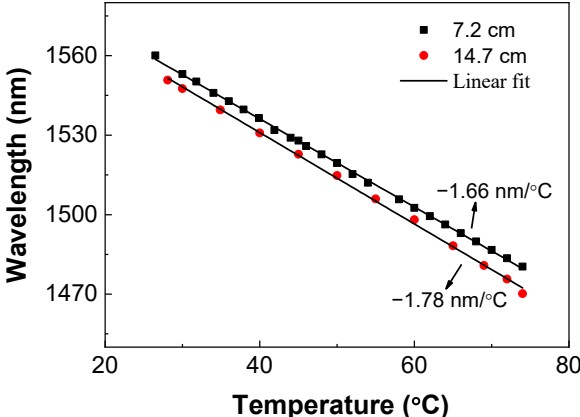

**Figure 7.** Wavelengths variation of dips for the fibers with lengths of 7.2 and 14.7 cm.

The transmission spectra of the boron-doped PM-PCF with 14.7 cm length at 30, 35 and 40 °C are shown in Figure 8a. In order to quantitatively study the temperature sensitivity, the wavelength values of the dips A to F at the temperature range of 28–76 °C were extracted (Figure 8b). The transmissions of dips A to F increased with temperature and shifted linearly blue. The temperature sensitivity of wavelength shift was obtained by linear fitting, showing values of −1.62~−1.83 nm/°C within the temperature range of 28~76 °C. The experimental temperature sensitivities were a little higher than the simulated value of −1.2 nm/°C. The temperature sensitivity in the long-wavelength direction is slightly higher, which is consistent with the observation of a more significant dip shift in the long-wavelength interference. The linear blue shift of the transmission dip observed in the experiment was coincident with the results of numerical analysis that showed similar S values at different temperatures. The temperature sensitivity of the designed PM-PCF is comparable to that reported in previous works. For example, Kaczmarek arrives at a temperature sensitivity of −1.7 nm/°C using a panda-type PMF SI [16]. Shao et al. [15] obtained a high-sensitivity temperature of 1.73 nm/°C by bending a spliced fiber of standard single mode fibers–PMF–single mode fibers into a circle of 30 mm. Recently, Zhao et al. [17] developed a PM-PCF filled with glycerin to achieve high sensitivity temperature of 1.5005 nm/°C at −25–85 °C. These findings indicate that our designed PM-PCF could be used as a good temperature sensor with high sensitivity and a wide temperature range.

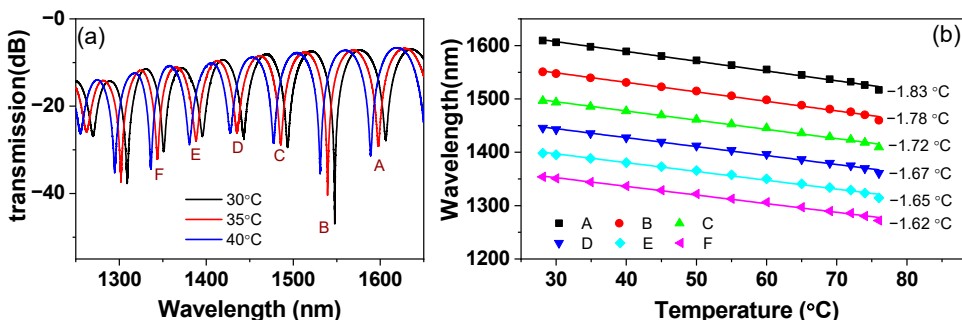

**Figure 8.** (**a**) Transmission spectra of PM-PCF-based SI at 14.7 cm length, and (**b**) wavelength variations (the discrete points) of dips A-F at the temperature range of 28–76 °C, where the solid curves are the linear fitting curves of experimental results.

Although there are deviations between the simulated and the experimental results, the former results are still in line with the experimental phenomena and mechanisms. For example, the numerical analysis in the previous section indicated the temperature sensitivity of the designed sensor mainly comes from the variation of birefringence with temperature, and the wavelength drift caused by the thermal expansion of the fiber length can be ignored. For the boron-doped PM-PCF, the birefringence mainly comes from the stress

birefringence introduced by the boron rods of SAPs. Thus, the good temperature sensitivity of the boron-doped PM-PCF SI temperature sensor depends on the stress birefringence introduced by the boron rod, which is sensitive to temperature changes.

## 5. Conclusions

In this paper, a boron-doped PM-PCFs-based Sagnac interferometer was developed for the temperature sensor. We theoretically calculated the birefringence characteristics of boron-doped PM-PCF, which showed that the fiber birefringence was mainly derived from the stress birefringence. The temperature sensitivies of the boron-doped PM-PCF SI were calculated to be around $-1.2$ nm/$°C$. Using the fabricated boron-doped PM-PCF SI, we found that fiber length had little impact on the temperature sensitivity. The highest temperature sensitivity of $-1.83$ nm/$°C$ was achieved within the temperature range of 28~76 °C. The theoretical and experimental studies verified that the temperature sensitivity of boron-doped PM-PCF SI depends on the stress birefringence of boron-doped stress-applying parts. Taken together, these findings suggest that the Sagnac interferometer with boron-doped PM-PCF has high temperature measuring sensitivity and a large measuring range, which depicts significant application prospects. Although a relatively high sensitivity of temperature was obtained by the special design, the temperature sensitivity of the proposed sensor could be further improved by filling a liquid or other materials.

**Author Contributions:** Conceptualization, L.C.; methodology, L.C.; software, L.C. and J.L.; validation, L.C., J.L. and S.X.; formal analysis, L.C.; investigation, L.C.; resources, L.C., J.L. and S.X.; data curation, L.C. and S.X.; writing—original draft preparation, L.C. and S.X.; writing—review and editing, L.C., S.X. and Y.T.; visualization, L.C. and S.X.; supervision, Y.T.; project administration, L.C. and Y.T.; funding acquisition, L.C. All authors have read and agreed to the published version of the manuscript.

**Funding:** This research was funded by Guidance Project of Science and Technology Research Program of Hubei Provincial Department of Education (No. B2021338, and B2016385), and the Scientific Research Foundation for Doctor of Wenhua College (No. 2019Y09), and School-level Scientific Research Project of Wenhua College (No. 2020Y08).

**Data Availability Statement:** Not applicable.

**Acknowledgments:** We appreciate the technical and materials support provided by the FLTG group under the leadership of Jinyan Li from Huazhong University of Science and Technology.

**Conflicts of Interest:** The authors declare no conflict of interest.

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
