# Peer review of "Sagnac Interferometric Temperature Sensor Based on Boron-Doped Polarization-Maintaining Photonic Crystal Fibers"

_optics, doi:10.3390/opt3040034_

Round 1

Author Response

We appreciate the reviewer’s high recognition of our study. We are also grateful for the reviewer’s insightful comments regarding a lot of work done in this paper. The very positive feedback from the reviewer motivated us to invest further efforts to improve the quality of manuscript and to render it suitable for publication in Optics. We have provided detailed responses and clarification in response to the reviewer’s comments in an attached file.

Reviewer 2 Report

1.      Explanation of the abbreviation (SIs) is given twice (line 24 and 29).

2.      Instead of the term "quartz glass", authors usually use "silica glass" in articles.

3.      On line 32: "silica glass" must be used instead of "quartz", since only the glassy state of SiO2 has a low thermal expansion coefficient.

4.      Remark on Equations 5 and 6: Birefringence in boron-containing fibers is created by elastic and hydrostatic mechanisms. (Ourmazd A., Malkolm P., Varnham M. P., Birch R. D., Payne D. N. Thermal properties of highly birefringent optical fibers and performs // Appl. Оpt. 1983. V. 22.  â„–15. P. 2374–2379).

5.       In fig. 7 in three places it is necessary to insert "○С".

6.      References [5], [7] and [9] do not contain the journal name.

Author Response

  1. Explanation of the abbreviation (SIs) is given twice (line 24 and 29).

Thanks. The extra explanation of the abbreviation (SIs) was removed from the revised manuscript.

  1. Instead of the term "quartz glass", authors usually use "silica glass" in articles.

Thanks. The term "quartz glass" was replaced with "silica glass" throughout the revised manuscript.

  1. On line 32: "silica glass" must be used instead of "quartz", since only the glassy state of SiO2has a low thermal expansion coefficient.

Thanks. The term "quartz glass" was replaced with "silica glass" throughout the revised manuscript.

  1. Remark on Equations 5 and 6: Birefringence in boron-containing fibers is created by elastic and hydrostatic mechanisms.

Thanks for the great suggestion. We have added a sentence to descript the birefringence in boron-containing fibers in the explanation of Equations 5 and 6.

Lines 100-109: For this highly birefringent fiber containing SAP, its modal birefringence can be divided into three parts [12]:

,                          (5)

where  is the birefringence caused by the geometric asymmetry of the core and cladding,  is the stress birefringence caused by SAPs, and  is the self-stress birefringence caused by the thermal expansion difference of the core. Birefringence in boron-containing fibers is created by elastic and hydrostatic mechanisms [13]. Since the core of boron-doped PM-PCF is pure quartz, the item . That is, modal birefringence simplifies to:

.                              (6)

Reference

[13] A. Ourmazd, M.P. Varnham, R.D. Birch, D.N. Payne, Thermal properties of highly birefringent optical fibers and preforms, Appl. Opt. 22 (1983) 2374-2379, https://10.1364/AO.22.002374.

  1. In fig. 7 in three places it is necessary to insert "○С".

Thanks for the great suggestion. The Fig.7 have been modified in the revised manuscript, and also copied here as Figure. R2.

Figure R2. Wavelengths variation of dips for the fibers with lengths of 7.2 cm and 14.7 cm.

  1. References [5], [7] and [9] do not contain the journal name.

Thanks. The references of [5], [7] and [9] were corrected as follows. We also checked all the references in the revised manuscript.

[5] J.C. Knight, Photonic crystal fibres, Nature 424 (2003) 847-851, https://doi.org/10.1038/nature01940.

[7] A. Ortigosa-Blanch, J.C. Knight, W.J. Wadsworth, J. Arriaga, B.J. Mangan, T.A. Birks, P.S.J. Russell, Highly birefringent photonic crystal fibers, Opt. Lett. 25 (2000) 1325-1327, https://doi.org/10.1364/OL.25.001325.

[9] Z. Chun-Liu, Y. Xiufeng, L. Chao, J. Wei, M.S. Demokan, Temperature-insensitive Interferometer using a highly birefringent photonic Crystal fiber loop mirror, IEEE Photonics Technol. Lett. 16 (2004) 2535-2537, https://doi.org/10.1109/LPT.2004.835646.

Round 2

Reviewer 1 Report

Overall, the authors have replied to my questions and recommendations in detailed. I am happy to make a recommendation of this manuscript in Optics.

 Before the paper being published, I would still recommend the authors take some time to submit a version of the manuscript that is final, only the changes are highlighted. There are many online and offline tools for comparing the old and new PDF, including "draftable" and etc.

Author Response

Thanks for the great suggestions. We have provided a final version with the changes marked, using the tool of "draftable".